# Exosome-mediated apoptosis pathway during WSSV infection in crustacean mud crab

Yi Gong[1,2,3,4], Tongtong Kong[1,2,3], Xin Ren[1,2,3], Jiao Chen[1,2,3], Shanmeng Lin[1,2,3], Yueling Zhang[1,2,3,4], Shengkang Li[1,2,3,4]*

1 Guangdong Provincial Key Laboratory of Marine Biology, Shantou University, Shantou, China, 2 Institute of Marine Sciences, Shantou University, Shantou, China, 3 Southern Marine Science and Engineering Guangdong Laboratory, Guangzhou, China, 4 STU-UMT Joint Shellfish Research Laboratory, Shantou University, Shantou, China

* lisk@stu.edu.cn

**Data Availability Statement:** Exosomal miRNAs seq data is available via the BioProject accession number PRJNA600674.

## Abstract

MicroRNAs are regulatory molecules that can be packaged into exosomes to modulate cellular response of recipients. While the role of exosomes during viral infection is beginning to be appreciated, the involvement of exosomal miRNAs in immunoregulation in invertebrates has not been addressed. Here, we observed that exosomes released from WSSV-injected mud crabs could suppress viral replication by inducing apoptosis of hemocytes. Besides, miR-137 and miR-7847 were found to be less packaged in mud crab exosomes during viral infection, with both miR-137 and miR-7847 shown to negatively regulate apoptosis by targeting the apoptosis-inducing factor (AIF). Our data also revealed that AIF translocated to the nucleus to induce DNA fragmentation, and could competitively bind to HSP70 to disintegrate the HSP70-Bax (Bcl-2-associated X protein) complex, thereby activating the mitochondria apoptosis pathway by freeing Bax. The present finding therefore provides a novel mechanism that underlies the crosstalk between exosomal miRNAs and apoptosis pathway in innate immune response in invertebrates.

## Author summary

As a form of intercellular vesicular transport, exosomes are widely involved in the regulation of a variety of pathological processes in mammals, yet, the role of exosomes during virus infection in crustaceans remains unknown. In the present study, we identified the miRNAs packaged by exosomes that were possibly involved in WSSV infection by mediating hemocytes apoptosis in crustacean mud crab *Scylla paramamosain*. The results revealed that exosomes released from WSSV-injected mud crabs could suppress viral replication by inducing hemocytes apoptosis. Moreover, it was found that miR-137 and miR-7847 were less packaged in exosomes after WSSV challenge, resulting in the activation of AIF, while AIF could translocate to nucleus to induce DNA fragmentation or disintegrate the HSP70-Bax complex and freeing Bax to mitochondria, which eventually caused apoptosis and suppressed viral infection of the recipient hemocytes. Our finding is the first to reveal the involvement of exosomal miRNAs in antiviral immune response in mud crabs,

**Funding:** This study was financially supported by the National Natural Science Foundation of China (31802341, 41876152), Key Special Project for Introduced Talents Team of Southern Marine Science and Engineering Guangdong Laboratory (Guangzhou) (GML2019ZD0606), Natural Science Foundation of Guangdong Province, China (2018A030307044), Department of Education of Guangdong Province, China (2017KQNCX072), STU Scientific Research Foundation for Talents (NTF18001), and Guangdong Provincial Special Fund for Modern Agriculture Industry Technology Innovation Teams (2019KJ141). The funders had no role in study design, data collection and analysis, decision to publish, or preparation of the manuscript.

**Competing interests:** The authors declare no conflicts of interest.

which shows a novel molecular mechanism of invertebrate resistance to pathogenic microbial infection.

## Introduction

Exosomes (measuring 30–120 nm in diameter) are extracellular vesicles of endocytic origin that are released into the extracellular environment under physiological and pathological conditions [1, 2]. They can be produced by various types of donor cells and then transferred to target cells, which serve as mediators during intercellular communications by transporting information cargo, including lipids, proteins, mRNAs and microRNAs (miRNAs) [3, 4]. Specific proteins highly enriched in exosomes are usually used as exosomes markers, such as TSG101, CD9, CD63 and CD81 [5, 6]. As a form of intercellular vesicular transport, exosomes are involved in the regulation of a variety of pathological processes [7]. Recently, exosomes have been implicated in viral pathogenesis and immune responses [8, 9]. During viral infection, the infected host cells can excrete exosomes containing viral or host genetic elements to neighboring cells to help modulate host immune response [10, 11], which suggest a crucial role of exosomes during viral infection. However, very little is known about how exosomes regulate host immune response and impact on viral infection.

miRNAs are small non-coding RNAs of 18–25 nucleotides in length that can bind to the 3'-untranslated region (UTR) of target genes in most cell types [12, 13]. Binding of the miRNAs can lead to recruitment of the target mRNAs to RNA-induced silencing complex (RISC), which result in translational arrest or mRNA degradation and decreased protein expression of the target genes [14, 15]. Apart from their endogenous actions, miRNAs can be secreted into the extracellular space within exosomes, with these miRNA-containing exosomes being taken up into neighboring or distant cells to modulate the expression of multiple target genes in the recipient cells [16, 17]. RNA sequencing analysis has shown that miRNAs are the most abundant among exosomal RNA species [18]. Recent evidences indicate that the alteration of miRNA composition can significantly affect the biological activities of exosomes that have been taken-up during viral infection [19, 20]. Importantly, it has been demonstrated that packaging of miRNAs into exosomes is selective and can also reflect dysregulated miRNA composition in donor cells [21]. It is conceivable that exosome-mediated intercellular transfer of miRNAs contribute to immune defense of the recipient cells and regulate viral spread.

White spot syndrome virus (WSSV) is a large enveloped double-stranded DNA virus that is very lethal and causes huge economic losses in aquaculture [22]. Studies have shown the widespread pathogenicity of WSSV among many marine crustaceans, including shrimp, crayfish and crab [23]. Invertebrates use innate immune responses (humoral and cellular) to recognize and protect themselves against pathogenic microbes [24]. Apoptosis is one type of cellular immune response that plays an essential role in host antiviral immunity [25], with viral infection capable of inducing apoptosis in infected cells in both vertebrates and invertebrates [26]. Given that exosomes are widely thought to be effective host antiviral defense tools [27], while marine crustaceans possess an open circulatory system, makes this open circulatory system an ideal carrier for exosomes to perform their immune functions. However, the role of exosomes during antiviral immune response in marine crustaceans is uncler, and the involvement of apoptosis remains unknown.

In an attempt to explore the involvement of exosomes in apoptosis during antiviral immunoregulation of marine crustaceans, the exosomal miRNAs that are potentially involved in WSSV infection were characterized in mud crab *Scylla paramamosain*. The results revealed

that exosomes released from WSSV-injected mud crabs could suppress viral replication by inducing hemocytes apoptosis. Moreover, it was found that miR-137 and miR-7847 were less packaged in exosomes after WSSV challenge, resulting in the activation of AIF (apoptosis-inducing factor), which eventually caused apoptosis and suppressed viral infection of the recipient hemocytes.

## Results

### The involvement of exosomes in antiviral regulation of mud crab

To characterize exosomes from mud crab during WSSV infection, exosomes isolated from the hemolymph of PBS-injected (i.e., exosome-PBS) and WSSV-injected (i.e., exosome-WSSV) mud crabs were used. The cup-shaped structure and size of the isolated exosomes were detected by electron microscopy (Fig 1A) and Nanosight particle tracking analysis (Fig 1B). In addition, Western blot analysis of exosome markers (CD9 and TSG101) and cytoplasmic marker (calnexin) were used to further ascertain that the isolated particles were indeed

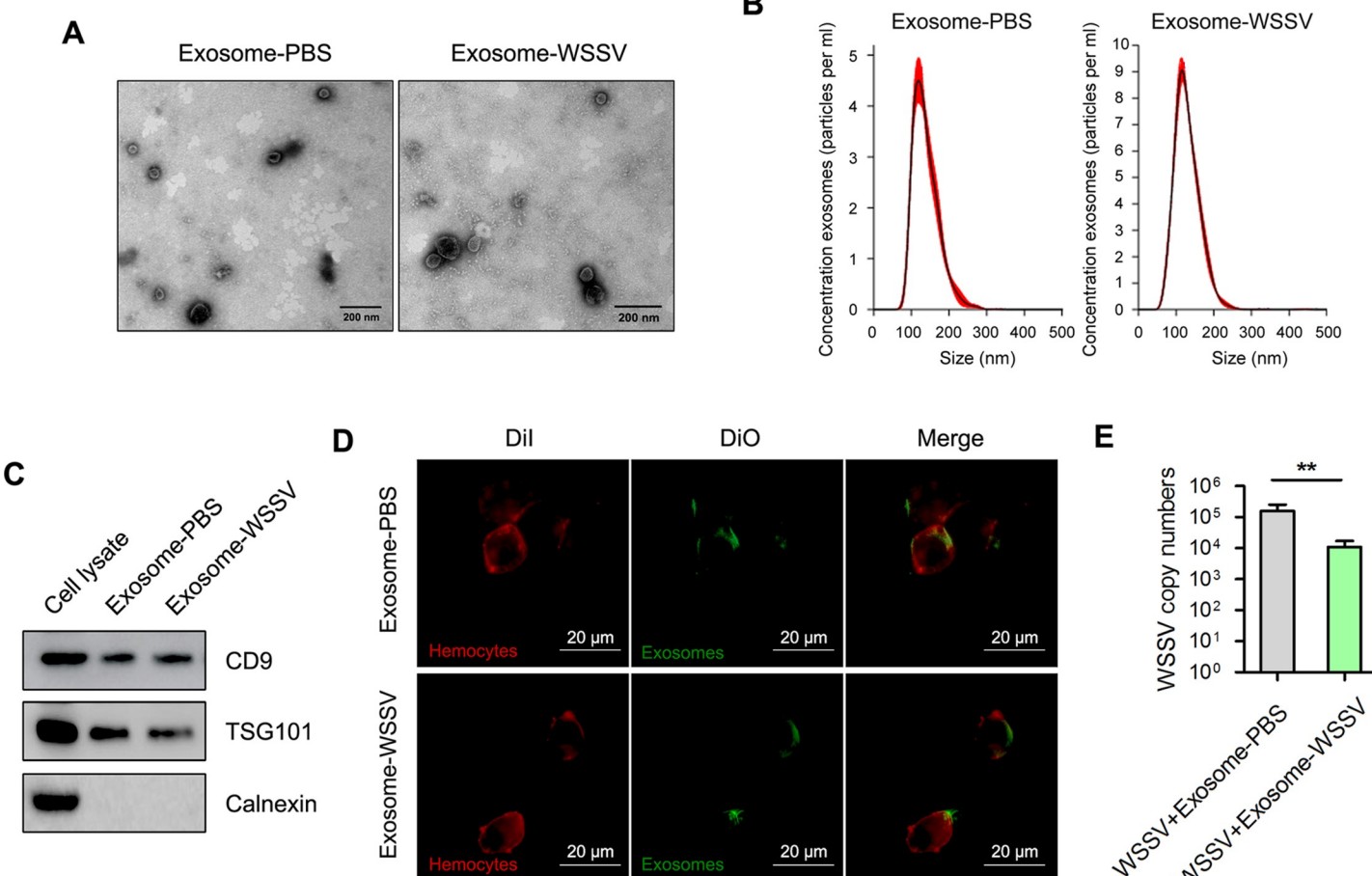

**Fig 1. Exosomes secreted from WSSV-infected mud crab participate in antiviral regulation. (A-B)** Exosomes isolated from mud crab with different treatments were detected by electron microscopy **(A)** and Nanosight particle tracking analysis **(B)**. Scale bar, 200 nm. **(C)** Western blotting assay of exosomal marker proteins (CD9 and TSG101) and cytoplasmic marker protein Calnexin in cell lysate and exosomes. **(D)** Confocal imaging showed the delivery of Dio-labeled exosomes (green) to Dil-labeled mud crab hemocytes (red). Scale bar, 20 μm. The indicated exosomes were injected into mud crab for 6 h, then the hemocytes were isolated and subjected to confocal imaging analysis. **(E)** The involvement of exosomes during WSSV infection, mud crabs were co-injected with the indicated exosomes and WSSV for 48 h, followed by the detection of WSSV copies. All data represented were the mean ± s.d. of three independent experiments (**, $p < 0.01$).

exosomes (Fig 1C). Furthermore, to analyze the capacity of the isolated exosomes to be internalized by hemocytes, mud crabs were injected with exosomes labeled with DiO (green), after which hemocytes were isolated and labeled with DiI (red). Confocal microscopic observation showed that the isolated exosomes (from PBS- and WSSV-injected mud crabs) were successfully internalized in hemocytes (Fig 1D). Besides, the effects of these exosomes on WSSV replication was determined using real-time PCR analysis. The results revealed that mud crabs injected with exosome-PBS had significantly higher WSSV copy number compared with exosome-WSSV injected crabs (Fig 1E). These results suggest that the secreted exosomes could be internalized in mud crab hemocytes, thereby playing an important role in the antiviral immune response of crabs.

## Exosome-mediated viral suppression is relevant for apoptosis activation

To investigate the involvement of apoptosis during exosome-mediated virus suppression, exosome-PBS or exosome-WSSV were co-injected with WSSV into mud crabs. At 48 hpi, Annexin V/PI staining followed by flow cytometric analysis revealed a higher number of apoptotic hemocytes in the exosome-WSSV and WSSV co-injected group compared with the other groups (control and exosome-PBS groups) (Fig 2A). A significant increase in Caspase 3/7 activity was also observed in the hemocytes of mud crabs co-injected with exosome-WSSV and WSSV compared with the control group (Fig 2B). To better understand the role of exosomes in mediating mitochondrial membrane potential, mud crabs were co-injected with either exosome-PBS or exosome-WSSV and WSSV, and hemocytes were analyzed using confocal microscopy. The confocal microscopic images revealed mitochondrial membrane potential with weak red fluorescence (based on JC-1 aggregates) and strong green fluorescence (JC-1 monomers) in both exosome-injected groups, compared with the controls (Fig 2C). Moreover, the levels of pro-apoptotic proteins, BAX and p53 increased, while levels of pro-survival proteins Bcl-2 and BAG1 were decreased in hemocytes of exosome co-injected mud crabs compared with controls (Fig 2D). To reveal the relationship between hemocytes apoptosis and virus infection in mud crab, the apoptosis inducer cycloheximide and apoptosis inhibitor Z-VAD-FMK were injected into crabs to examine their effect on WSSV replication in hemocytes. The results revealed significantly lower WSSV copy number in the cycloheximide and WSSV-injected group, while the Z-VAD-FMK and WSSV-injected group had significantly higher WSSV copy number, compared with the WSSV-injected group (Fig 2E), which suggest a negative role of apoptosis during virus infection. To further investigate the effect of apoptosis on exosome-mediated virus suppression, mud crabs were co-injected with WSSV and either exosome-PBS, exosome-WSSV, or exosome-WSSV and Z-VAD-FMK. It was found that the exosome-WSSV-mediated virus suppression was significantly reduced when apoptosis was inhibited by Z-VAD-FMK (Fig 2F). These results revealed that exosomes isolated from WSSV challenged mud crabs suppressed viral infection through apoptosis activation.

## Functional miRNA screening in exosomes

Microarray analysis of exosomal miRNAs was carried out with a 1.5-fold change and *P<0.01* used as threshold cut-off. The results showed that 124 miRNAs were differentially expressed in isolated exosomes released from the exosome-WSSV group compared with the exosome-PBS group. Among the differentially expressed miRNAs, 84 were upregulated and 40 were downregulated (see heatmap in Fig 3A). The top 10 differentially expressed miRNAs, which include miR-137, miR-60, miR-373, miR-7847, miR-87a, miR-513, miR-353, miR-81, miR-508, and miR-387, are shown in Fig 3A. To investigate the role of these miRNAs in WSSV replication, miRNA mimics and anti-miRNA oligonucleotides (AMOs) were synthesized and co-injected

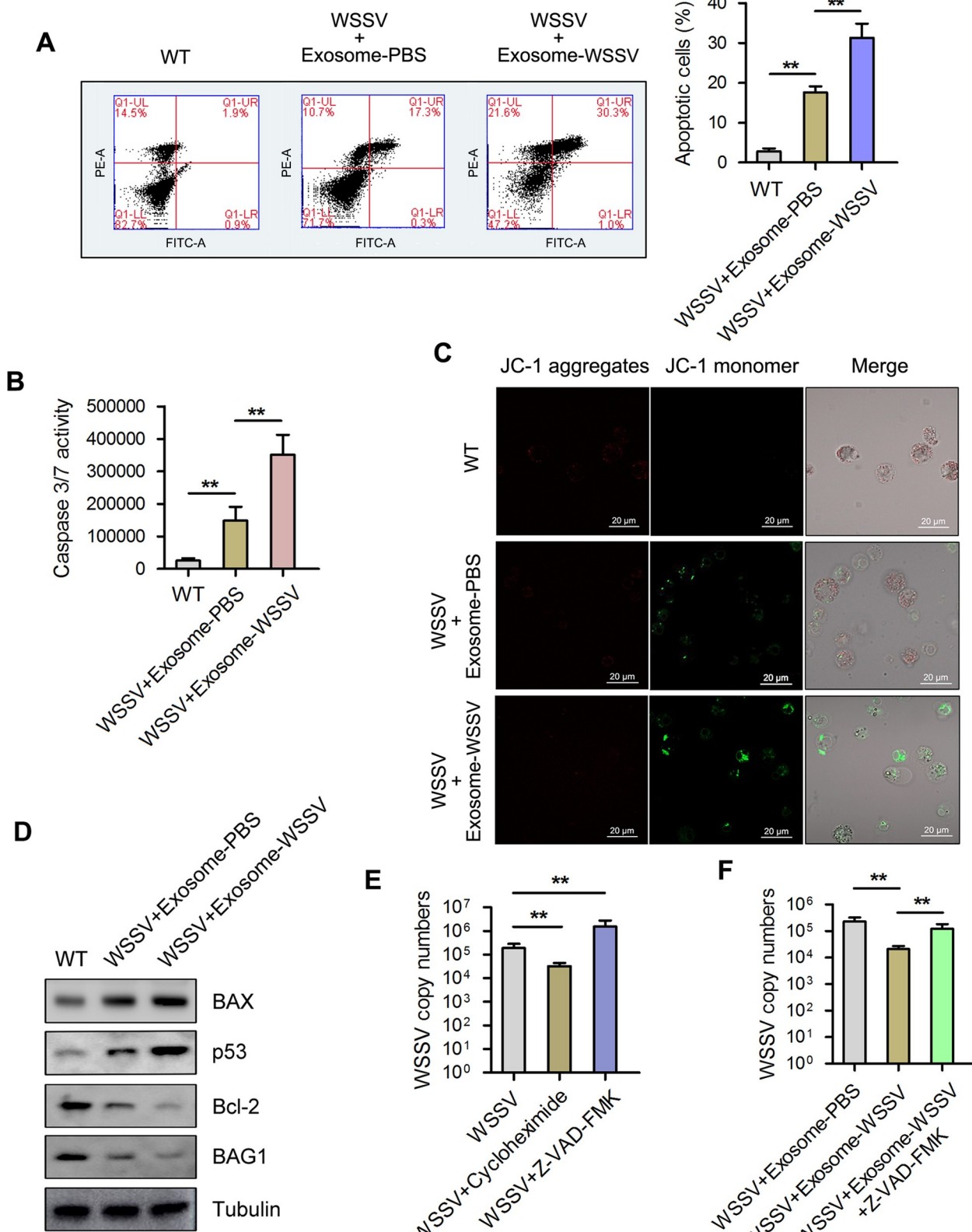

**Fig 2. Exosomes isolated from mud crab challenged with WSSV suppressed virus infection through activation of apoptosis. (A-D)** The influence of the indicated exosomes on apoptosis of mud crab hemocytes. The isolated exosomes from mud crab with different treatments

(including PBS and WSSV) were co-injected with WSSV into mud crab for 48 h, then, the apoptotic levels of the hemocytes were examined through annexin V assay **(A)**, caspase 3/7 activity analysis **(B)**, mitochondrial membrane potential measurement **(C)** and apoptosis-associated protein detection **(D)**. **(E)** The role of apoptosis regulation during virus invasion, apoptosis inducer cycloheximide or apoptosis inhibitor Z-VAD-FMK were co-injected into mud crab with WSSV for 48 h, followed by the detection of WSSV copies. **(F)** The involvement of apoptosis regulation during exosome-mediated virus suppression, mud crabs were co-injected with the indicated exosomes, WSSV and apoptosis inhibitor Z-VAD-FMK, then WSSV copy numbers were detected. All the data were the average from at least three independent experiments, mean ± s.d. (**, $p < 0.01$).

with WSSV into mud crabs for 48 h followed by qPCR analysis. The results revealed an increase in WSSV copy number in mud crabs after miRNA mimics injection and a decrease in WSSV copy number in mud crabs after AMO injection (Fig 3B and 3C). Next, the relative expression levels of miR-137 and miR-7847 were investigated in exosome-PBS and exosome-WSSV injected mud crabs. It was found that both miR-137 and miR-7847 were significantly downregulated in the exosome-WSSV injected group compared with the control group (Fig 3D).

In order to explore the roles of miR-137 and miR-7847 in apoptosis regulation, mud crabs were injected with AMO-miR-137 and AMO-miR-7847 followed by flow cytometric analysis of hemocytes. The results revealed that both AMO-miR-137 and AMO-miR-7847 induced higher hemocytes apoptosis (percentage of apoptotic cells), compared with the controls (WT and AMO-NC) (Fig 3E). Similarly, caspase 3/7 activity was significantly increased in AMO-miR-137 or AMO-miR-7847 injected groups, compared with the control groups (Fig 3F). Next, the participation of miR-137 and miR-7847 in exosome-mediated virus suppression was examine in mud crabs co-injected with either exosome-PBS, exosome-WSSV, exosome-WSSV and miR-137-mimic, or exosome-WSSV and miR-7847-mimic and WSSV. The WSSV copy number was significantly lower in the exosome-WSSV and WSSV co-injected group, compared with the other groups ($P < 0.05$) (Fig 3G). To explore the relationship between miR-137 and miR-7847, mud crabs were co-injected with either exosome-PBS, exosome-PBS and AMO-miR-137, exosome-PBS and AMO-miR-7847 and WSSV followed by WSSV copy number analysis. The data shows that separate silencing of miR-137 and miR-7847 could suppress viral replication (Fig 3H), which indicated that both miR-137 and miR-7847 could function separately during the exosomal regulatory process. Taken together, these results suggest that miR-137 and miR-7847 are controlled by WSSV-derived exosomes to promote viral replication in hemocytes of mud crabs.

## Interactions between miR-137 and miR-7847 with their targeted genes

To identify the pathways mediated by miR-137 and miR-7847 in mud crabs, their target genes were predicted using the Targetscan and miRanda softwares. The apoptosis-inducing factor (AIF) was predicted as the target gene of both miR-137 and miR-7847 (Fig 4A). To ascertain this prediction, synthesized miR-137 and miR-7847 as well as EGFP-AIF-3'UTR-miR-137/-miR-7847 or mutant plasmids (EGFP-ΔAIF-3'UTR-miR-137/-miR-7847) were co-transfected into *Drosophila* S2 cells (Fig 4B) followed by fluorescence microscopy. The results revealed that the fluorescence intensity in cells co-transfected with EGFP-AIF-3'UTR-miR-137 or EGFP-AIF-3'UTR-miR-7847 was significantly decreased compared with cells co-transfected with EGFP-ΔAIF-3'UTR-miR-137 or EGFP-ΔAIF-3'UTR-miR-7847, respectively (Fig 4C). This suggest that miR-137 and miR-7847 could potentially interact with AIF to modulate its expression.

In order to confirm the role of miR-137 and miR-7847 in the expression of AIF mRNA, the expression of miR-137 and miR-7847 were silenced using AMOs or miRNA mimics and analyzed by qPCR. The qPCR results showed that AIF transcripts were significantly increased following AMO-miR-137 or AMO-miR-7847 treatment, but was significantly decreased in the

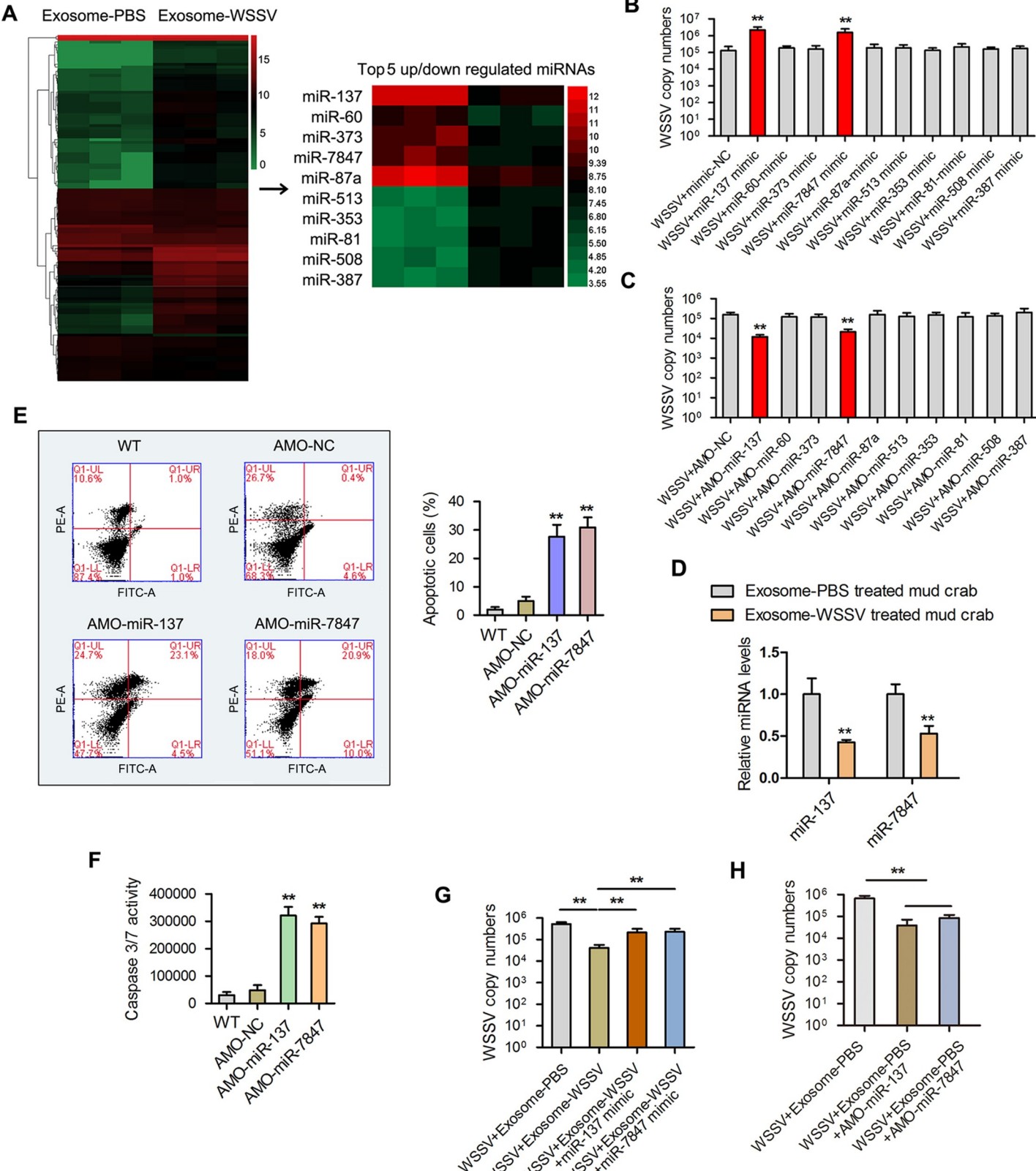

**Fig 3. Exosomal miR-137 and miR-7847 were characteristically secreted to mediate apoptosis and virus invasion in mud crab. (A)** Microarray analysis of exosomal miRNAs were presented in a heatmap, the top5 up/down regulated miRNAs in the indicated exosomes were listed in detail. **(B-C)** The effects of the indicated miRNAs

on virus infection, mimics or anti-miRNA oligonucleotides (AMOs) of the indicated miRNAs were co-injected with WSSV into mud crab for 48 h, then WSSV copy numbers were evaluated via qPCR. **(D)** The expression levels of miR-137 and miR-7847 in mud crab challenged with different exosomes. **(E-F)** The functions of miR-137 and miR-7847 on apoptosis regulation, AMO-miR-137 and AMO-miR-7847 were injected into mud crab separately, then the hemocytes were subjected to annexin V assay **(E)** and caspase 3/7 activity analysis **(F)**. **(G-H)** The participation of miR-137 and miR-7847 in exosome-mediated virus suppression. The indicated exosomes, WSSV, mimics or AMOs were co-injected into mud crabs, followed by the detection of WSSV copies using qPCR. Experiments were performed at least in triplicate and the data represented were the mean ± s.d. (**, $p < 0.01$).

miR-137-mimic-scrambled or miR-7847-mimic-scrambled groups, respectively, compared with control (Fig 4D and 4E). On the other hand, AMO-miR-137-scrambled or AMO-miR-7847-scrambled and miR-137-mimic or miR-7847-mimic had no significant effect on the expression of AIF mRNA (Fig 4D and 4E).

To investigate the targeting of AIF by miR-137 and miR-7847 in hemocytes of mud crab, co-localization of miR-137/miR-7847 and AIF mRNA was examined. Hemocytes were treated with FAM-labeled AIF mRNA probe (green), Cy3-labeled miRNA probe (red) and DAPI (blue) before been observed under a fluorescence microscope. The results showed that miR-137/miR-7847 and AIF mRNA were co-localized in the cytoplasm of cells (Fig 4F).

## Effects of AIF on WSSV proliferation and apoptosis

To ascertain whether AIF is involved in the immune response of mud crab, the expression profile of AIF was determined after WSSV challenge using Western blot and qPCR analyses. The results revealed that AIF was significantly elevated at 24 and 48 hours post-WSSV challenge (Fig 5A). In order to determine the effects of AIF on the proliferation of WSSV, viral copy number was examined in AIF depleted mud crabs challenged with WSSV. The results showed significantly higher WSSV copy number in AIF knockdown (siAIF-injected) mud crabs, compared with the control groups (GFP-siRNA and WSSV group) (Fig 5B).

In order to unravel the role of AIF in miR-137 and miR-7847-mediated apoptosis regulation, AIF-depleted mud crabs were injected with AMO-miR-137 or AMO-miR-7847 followed by flow cytometry and caspase-3/7 activity analyses in hemocytes. The results showed significantly higher percentage of apoptotic cells in untreated mud crabs injected with AMO-miR-137 or AMO-miR-7847 compared with AIF-depleted mud crabs injected with AMO-miR-137 or AMO-miR-7847, respectively (Fig 5C). Similar results were obtained for the caspase 3/7 activity analysis (Fig 5D). Moreover, AIF was found to participate in exosome-mediated virus suppression, as the expression of AIF was significantly increased in exosome-WSSV treated mud crabs compared with control (Fig 5E). In addition, AIF-depleted mud crabs co-injected with exosome-WSSV and WSSV had significantly higher WSSV copy number compared with normal mud crabs co-injected with either exosome-WSSV or exosome-PBS and WSSV (Fig 5F).

## Nuclear translocation of AIF induces DNA fragmentation

It has previously been reported that AIF is able to translocate into the nuclear of hemocytes [28]. Thus, Western blot analysis was used to determine whether AIF was present in the nuclear extract of hemocytes from mud crabs injected with WSSV, AMO-miR-137, AMO-miR-7847 or untreated. The results revealed that AIF was found in the nucleus of hemocytes in all groups (except untreated) at 6 and 24 hours post-injection (Fig 6A). The localization of AIF in mud crab hemocytes was further confirmed by immunofluorescence microscopy using mouse anti-AIF antibody. The results indicated co-staining of AIF with DAPI in the nuclei of hemocytes (Fig 6B). To explore the effect of AIF translocation to the nucleus, DNA fragmentation was analyzed on 3% agarose gel electrophoresis using genomic DNA isolated from

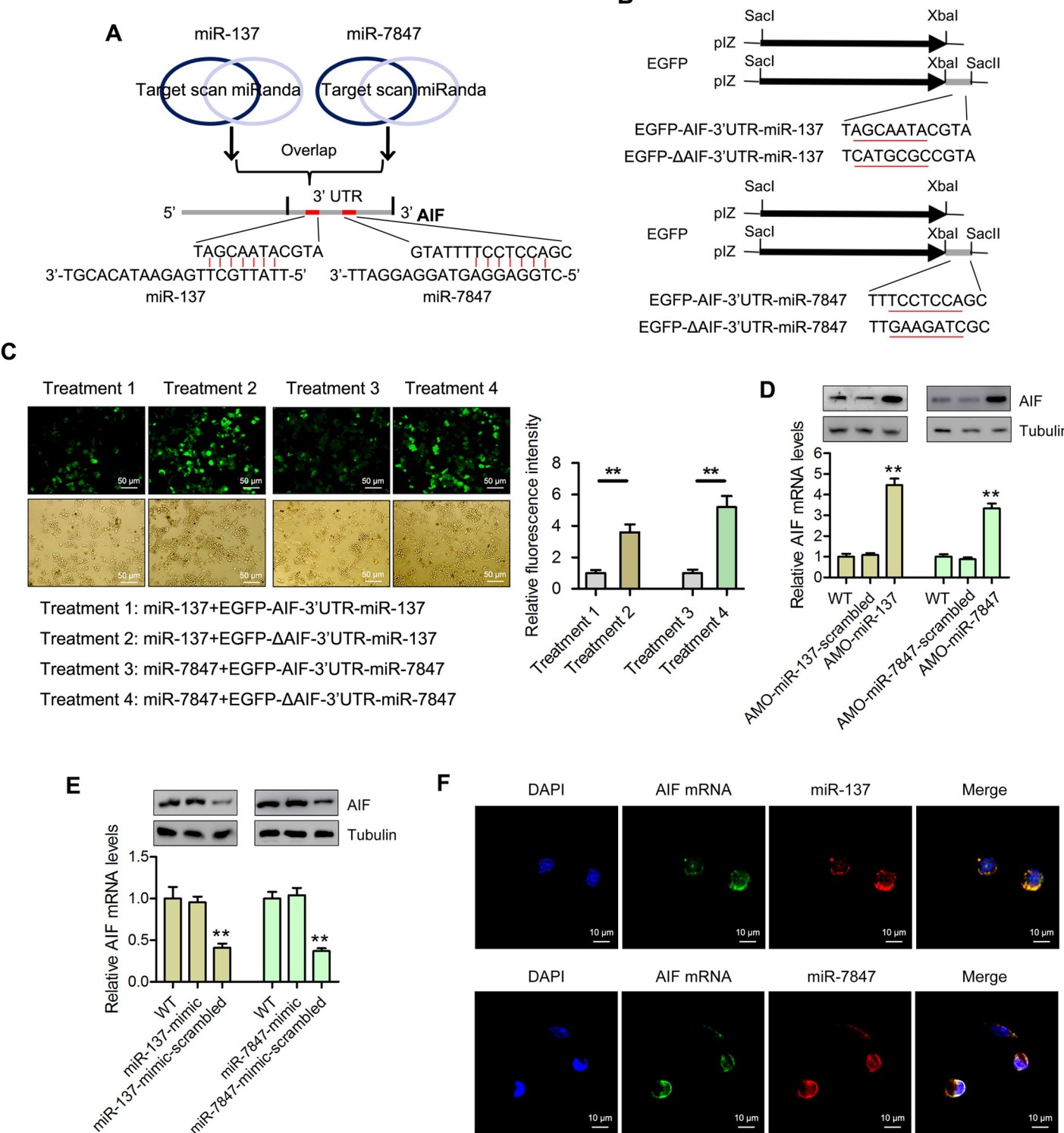

**Fig 4. AIF is a direct downstream target for both miR-137 and miR-7847 in mud crab. (A)** Target gene prediction of miR-137 and miR-7847 with two bioinformatics tools, as predicted, the 3'UTR of AIF could be simultaneously targeted by miR-137 and miR-7847. **(B)** The construction of the wild-type and mutated 3'UTRs of AIF. The sequences targeted by miR-137 and miR-7847 were underlined. **(C)** The direct interactions between miR-137, miR-7847 and AIF in insect cells, S2 cells were co-transfected with miR-137, miR-7847 and the indicated constructed plasmids for 48 h, then the relative fluorescence intensities were evaluated. **(D)** The effects of miR-137 and miR-7847 silencing on the expression levels of AIF in mud crab, AMO-miR-137 and AMO-miR-7847 were injected into mud crab separately, 48 h later, the mRNA and protein expression levels were examined. **(E)** The effects of miR-137 and miR-7847 overexpression on the mRNA and protein expression

levels in mud crab. **(F)** The co-localization of miR-137, miR-7847 and AIF mRNA in mud crab hemocytes, miR-137, miR-7847, AIF mRNA and nucleus of hemocytes were respectively detected with FAM-labeled AIF mRNA probe (green), Cy3-labeled miR-137 and miR-7847 probe (red) and DAPI (blue). Each experiment was performed in triplicate and data are presented as mean ± s.d. (**, $p<0.01$).

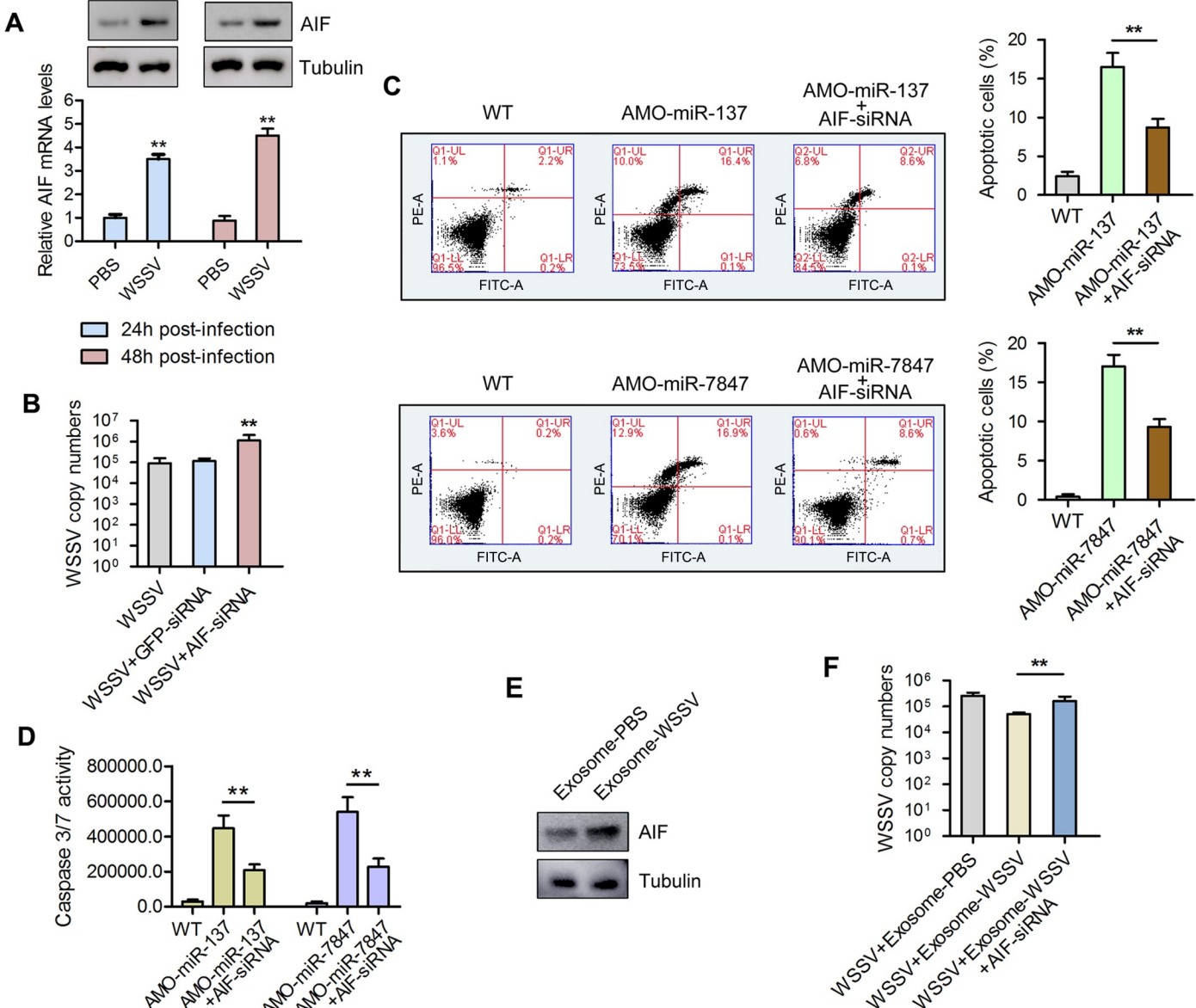

**Fig 5. Exosomal miR-137 and miR-7847 regulate apoptosis and virus invasion through targeting AIF in mud crab. (A)** The expression levels of AIF during WSSV infection, mud crabs treated with PBS or WSSV were subjected to western blot and qPCR analysis to detect the mRNA and protein levels of AIF. **(B)** The influence of AIF silencing on WSSV infection in mud crab. WSSV and AIF-siRNA were co-injected into mud crab for 48 h, followed by the detection of WSSV copy numbers, GFP-siRNA was used as control. **(C-D)** The involvement of AIF during miR-137 and miR-7847-mediated apoptosis regulation in mud crab. AMO-miR-137 and AMO-miR-7847 were co-injected with AIF -siRNA separately, then the hemocytes were subjected to annexin V assay **(C)** and caspase 3/7 activity analysis **(D)**. **(E)** The effect of the indicated exosomes on AIF expression. Isolated exosomes from mud crabs treated with PBS and WSSV were injected into mud crabs followed by determination of AIF protein level using Western blot. **(F)** The participation of AIF during exosome-mediated virus suppression, the indicated exosomes, WSSV and AIF-siRNA were co-injected into mud crabs, followed by the detection of WSSV copies. Data presented were representatives of three independent experiments (**, $p<0.01$).

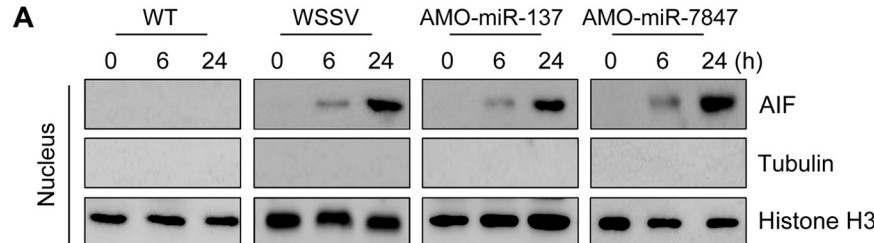

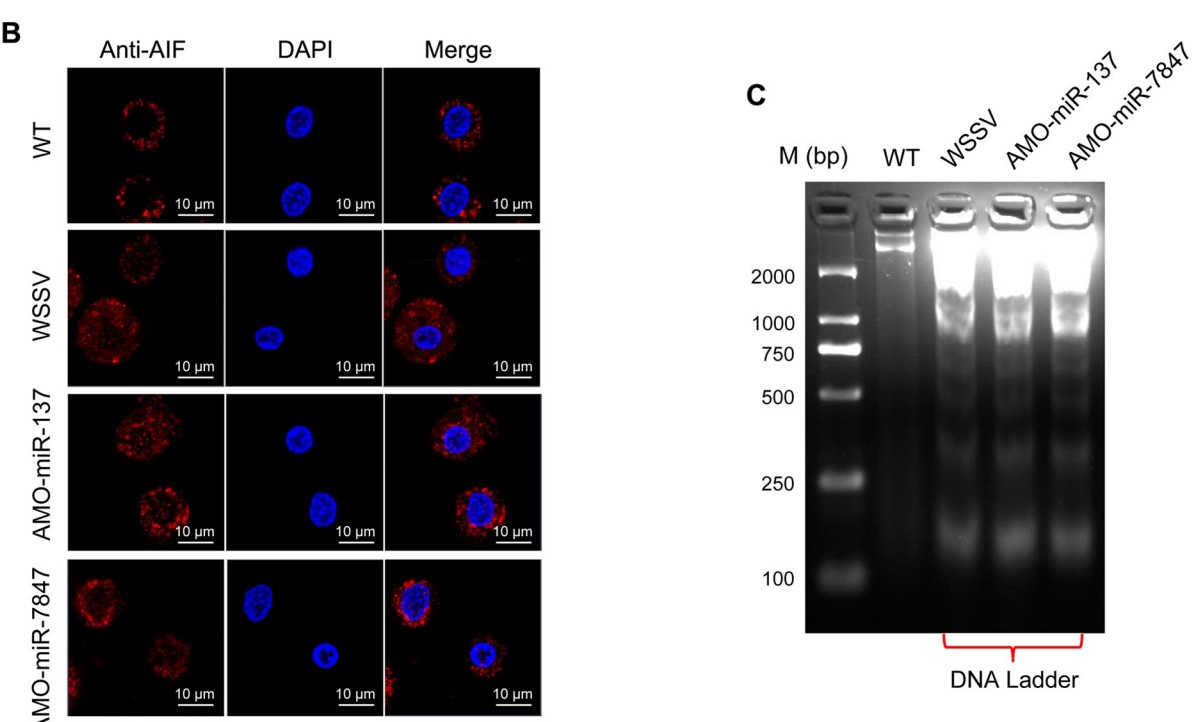

**Fig 6. AIF translocated into nucleus to mediate DNA fragmentation. (A)** The protein level of AIF in the nucleus, mud crabs were treated with WSSV, AMO-miR-137 and AMO-miR-7847 separately, at 0, 6 and 24 h after the treatments, the nucleus of hemocytes were isolated and subjected to western blot analysis. Tubulin and Histone H3 were used to evaluate the purity of the isolated nucleus. **(B)** Immunofluorescent assay was performed to detect the localization of AIF in mud crab hemocytes after specific treatment, including AMO-miR-137, AMO-miR-7847 and WSSV. Scale bar, 10 μm. **(C)** DNA Ladder detection of mud crab treated with AMO-miR-137, AMO-miR-7847 and WSSV separately, then the genomic DNA was isolated and subjected to agarose gel electrophoresis.

hemocytes of mud crabs injected with WSSV, AMO-miR-137, AMO-miR-7847 or untreated. The results revealed that WSSV, AMO-miR-137 and AMO-miR-7847 induced marked DNA fragmentation compared with the control group (Fig 6C).

## Role of AIF in mitochondrial apoptosis

Based on the observation that AIF was involved in hemocytes apoptosis, co-immunoprecipitation analysis was carried out followed by SDS-PAGE and Western blot analyses. The results indicated that AIF could bind to HSP70 (Fig 7A and 7B). To confirm the role of HSP70 in the regulation of apoptosis in hemocytes, mud crabs were depleted of HSP70 and the level of hemocytes apoptosis determined using flow cytometry. A significant percentage of apoptotic cells were were observed in hemocytes of HSP70 silenced mud crabs compared with the control groups (p<0.01) (Fig 7C). Knockdown of HSP70 also significantly decreased WSSV copy

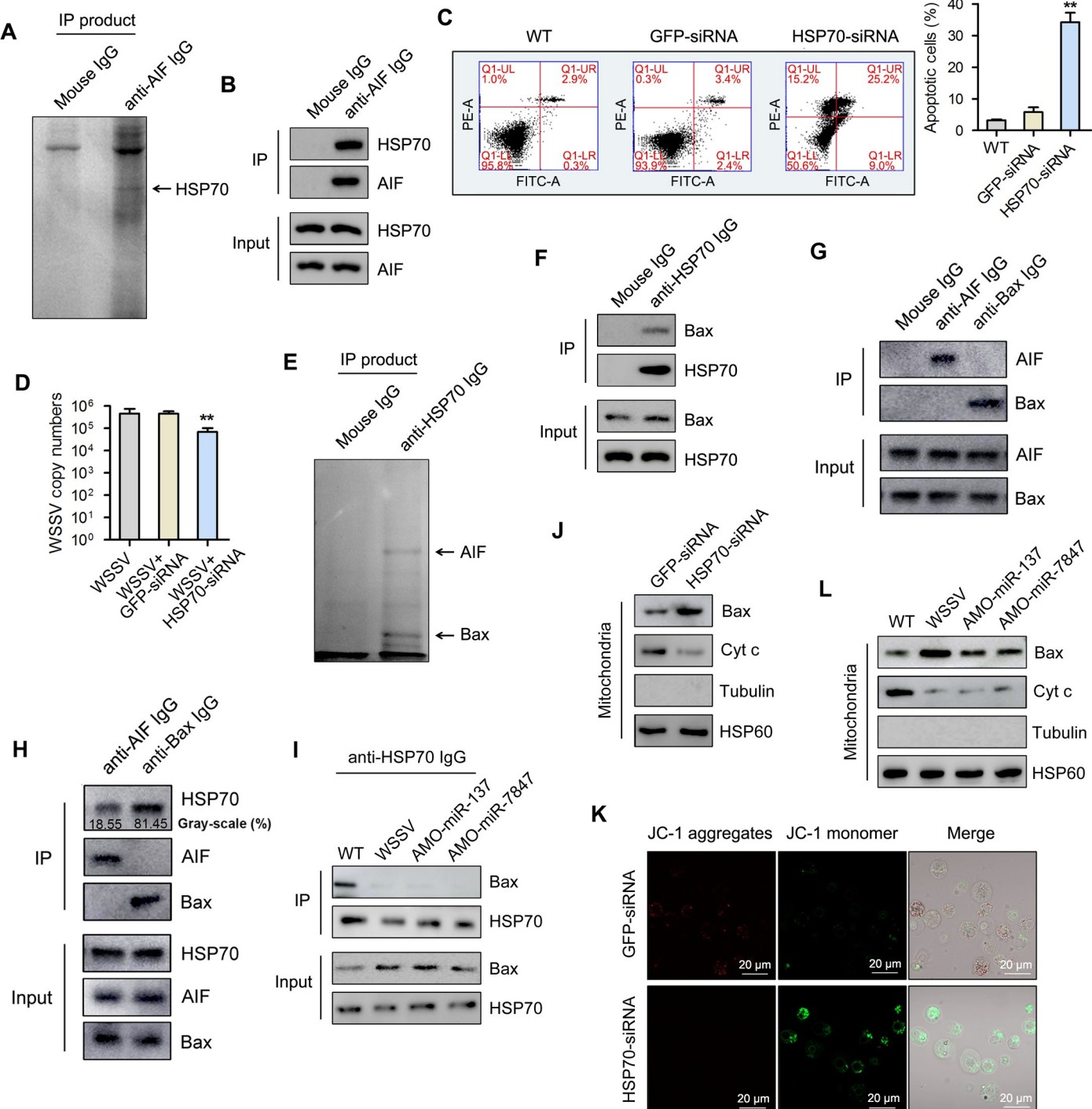

**Fig 7. Underlying mechanisms of AIF-mediated mitochondrial apoptosis process.** (**A**) Identification of proteins bound to AIF. The cell lysates of mud crab hemocytes were subjected to Co-IP assay using anti-AIF IgG, then the IP products were separated with SDS-PAGE and identified by mass spectrometry. (**B**) The interactions between AIF and HSP70 in mud crab, the cell lysates were subjected to Co-IP assay with anti-AIF IgG, then the IP products were subjected to Western blot analysis to detect HSP70. (**C**) The effects of HSP70 silencing on apoptosis regulation, HSP70-siRNA or GFP-siRNA were injected into mud crab for 48 h, then the hemocytes were subjected to annexin V assay. (**D**) The influence of HSP70 silencing on WSSV infection in mud crab. (**E**) Identification of proteins bound to HSP70, the proteins identified were indicated with arrows. (**F**) The interactions between HSP70 and Bax in mud crab, the cell lysates were subjected to Co-IP assay with anti-HSP70 IgG, then the IP products were subjected to Western blot analysis to detect Bax. (**G**) The interactions between AIF and Bax in mud crabs. Cell lysates were subjected to Co-IP assay with anti-AIF IgG and anti-Bax IgG, and the Co-IP products subjected to Western blot analysis to detect AIF and Bax. (**H**) The percent of HSP70 that binds to AIF and Bax. Cell lysates were subjected to Co-IP assay with anti-AIF IgG and anti-Bax IgG, followed by analyzing the Co-IP products by Western blot and Gray-scale sensitometry quantification using Image J software. (**I**) The interactions between HSP70 and Bax in mud crab with the

indicated treatments. **(J)** The influence of HSP70 silencing on the protein levels of Bax and Cyt c in mitochondria. **(K)** The effects of HSP70 silencing on mitochondrial apoptosis, HSP70-siRNA or GFP-siRNA were injected into mud crab for 48 h, then the hemocytes were subjected to mitochondrial membrane potential measurement. **(L)** The detections of Bax and Cyt c in mitochondria of mud crab with the indicated treatments. All the numeral data represented the mean ± s.d. of triplicate assays (**, $p < 0.01$).

number in hemocytes of mud crabs challenged with WSSV compared with the control groups (Fig 7D). This indicates the involvement of HSP70 in the replication of WSSV in mud crab hemocytes. Moreover, co-immunoprecipitation analysis revealed that HSP70 could bind to Bax (Fig 7E and 7F). Given that AIF could bind to HSP70, while HSP70 also binds to Bax, we went on to determine whether AIF could also directly bind to Bax using co-immunoprecipitation analysis. The results showed that AIF could not bind to Bax (Fig 7G), which seemed to suggest that HSP70 formed a complex with AIF and Bax separately. To analyze the percent of HSP70 that binds to AIF vis-à-vis Bax, co-immunoprecipitation analyses were performed with anti-AIF IgG and anti-Bax IgG, followed by measuring the percentage Gray-scale of the bands using Image J software. The results revealed 18.55% gray-scale for the HSP70—AIF interaction band and 81.45% for the HSP70—Bax interaction band (Fig 7H). Furthermore, the interaction between HSP70 and Bax is shown in Fig 7I. These results showed that the HSP70-Bax complex in hemocytes was disrupted when mud crabs were injected with either AMO-miR-137, AMO-miR-7847 or WSSV compared with control (WT group) (Fig 7I).

In the mitochondria of HSP70-depleted mud crabs, the expression of Bax was observed to increase, while that of Cyt C decreased (Fig 7J), which suggest a role of HSP70 in regulating the functions of Bax and Cyt C in mitochondria. To further explore whether silencing of HSP70 could affect mitochondrial mediated apoptosis, mitochondrial membrane potential of hemocytes from HSP70-silenced mud crabs was determined using confocal microscopy in terms of JC-1 aggregates (red fluorescence) and JC-1 monomers (green fluorescence). The results (Fig 7K) revealed weak red fluorescence and strong green fluorescence in the HSP70-depleted group compared with the control group, which indicates high apoptosis rate in HSP70 knockdown mud crabs. To detect the presence of both Bax and Cyt C in mitochondria, mud crabs were injected with either AMO-miR-137, AMO-miR-7847, WSSV or untreated followed by Western blot analysis. The results revealed an increased level of Bax and a decreased level of Cyt C in mitochondria of AMO-miR-137, AMO-miR-7847 or WSSV treated groups compared with control (Fig 7L).

Taken together, our findings revealed that during WSSV infection, there was less packaging of miR-137 and miR-7847 in mud crab exosomes, with the decreased uptake of exosomal miR-137 and miR-7847 resulting in AIF activation in the recipient hemocytes. While AIF could translocate to nucleus to induce DNA fragmentation, besides, it could also competitively bind to HSP70 to disintegrate the HSP70-Bax complex, then the freeing Bax was transferred to mitochondria, which eventually cause mitochondrial apoptosis and suppress viral infection in the recipient hemocytes (Fig 8).

## Discussion

Exosomes are small membrane-enclosed vesicles actively released by cells into the extracellular environment, with the molecular cargo carried by exosomes reflecting the physiological or pathological state of donor cells [29]. In recent years, the involvement of exosomes in viral pathogenesis and immune responses has been widely investigated [8]. For instance, exosomes can protect viral contents from immune recognition, with studies showing that exosomes secreted by hepatitis C virus (HCV)-infected cells contain full-length viral RNA, which can be

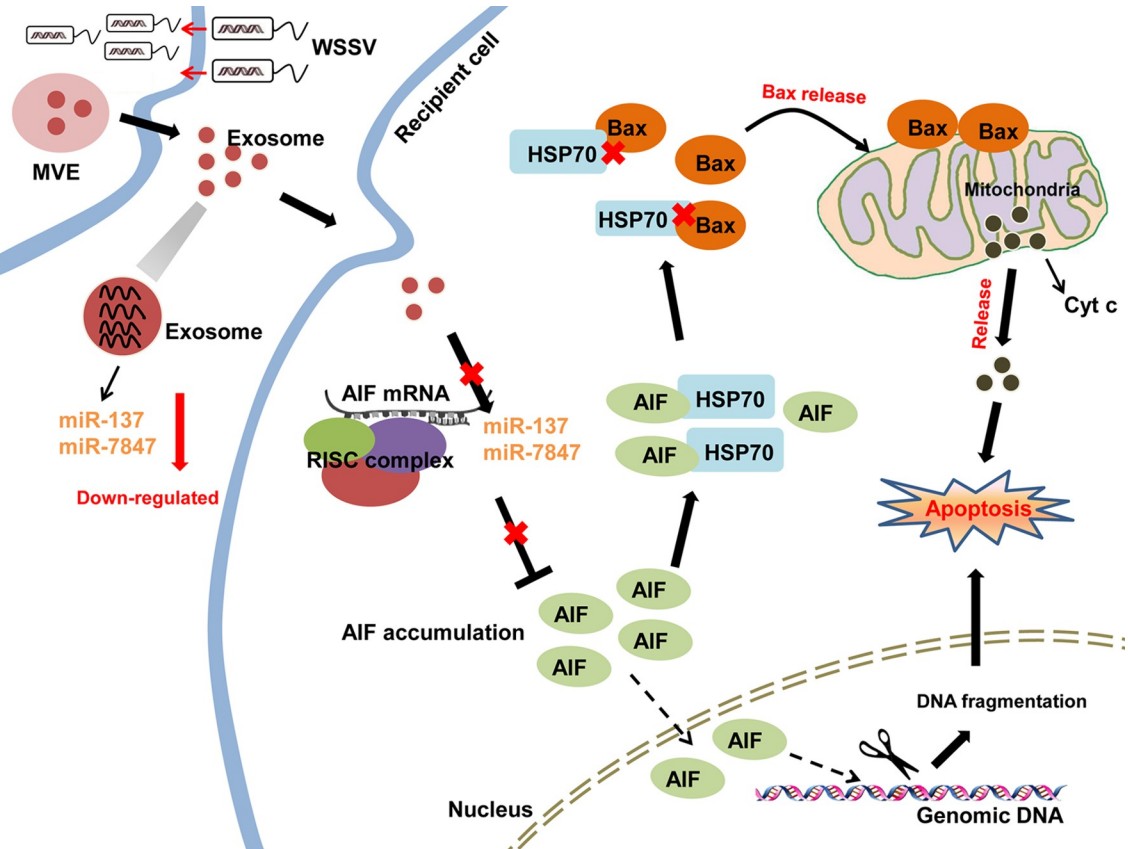

**Fig 8. Proposed schematic diagram for the exosomal miR-137 and miR-7847-mediated apoptosis and virus invasion regulation in mud crab.**

delivered to dendritic cells to establish infection [30, 31]. Similarly, VSV (vesicular stomatitis virus) infection can mediate the recruitment of TRAMP-like complex to exosomes to recognize and induce degradation of viral mRNAs [32]. In addition, exosomes released from HIV-infected cells contain regulatory factors required for apoptosis activation, which inhibit virus invasion by inducing apoptosis of uninfected cells [33]. Although there has been an increasing number of studies on exosomes and viral infection, most are focused on higher organisms, while the role of exosomes in antiviral immune response of invertebrates has largely been unexplored. In the current study, we revealed for the first time that, exosomes isolated from mud crab have typical characteristics as those in higher organisms. Furthermore, our data shows that exosomes released from WSSV-injected mud crabs could suppress viral replication by inducing hemocytes apoptosis, which demonstrates a novel role of exosomes in invertebrates.

A distinct feature of exosomes is that a large number of nucleic acids are packaged in it, including miRNA, mRNA, mtDNA, piRNA, 1ncRNA, rRNA, snRNA, snoRNA and tRNA [34]. As the most abundant RNA in exosomes, studies have repeatedly demonstrated the essential roles of exosomal miRNAs during host-virus interactions [35]. Besides, the miRNA cargo carried by exosomes can be affected by external signals such as oxidative stress and pathogens infection [36], which possess completely different molecular composition to deal with these stimulations. For instance, EV71 infection cause differential packaging of miR-146a to exosomes, which suppresses type I interferon expression in the recipient cells, thus facilitating

viral replication [37]. miR-483-3p is highly presented in mice exosomes during influenza virus infection, which potentiates the expression of type I interferon and proinflammatory cytokine genes [35]. Exosomal miR-145, miR-199a, miR-221 and Let-7f secreted by umbilical cord mesenchymal stem cells can directly bind to the genomic RNA of HCV and effectively inhibit viral replication [38]. In addition, studies have found that exosomal miRNAs are endowed with other functions apart from regulation of gene expression. Exosomal miR-21 and miR-29a secreted by HEK293 cells can serve as ligands and bind with toll-like receptors (TLRs), thus activating relevant immune pathways in the recipient cells [39]. Due to their diverse mode of regulation and functions, exosomal miRNAs are crucial regulators in response to virus infection, although no relevant research has been carried out in invertebrates. In the current study, we found that miR-137 and miR-7847 were less packaged in mud crab exosomes after WSSV challenge. Both miR-137 and miR-7847 are negative apoptosis regulators that target AIF, hence, decreased uptake of exosomal miR-137 and miR-7847 results in the activation of AIF, which eventually induced apoptosis and suppressed viral infection in recipient mud crab hemocytes. The present study thus reveals a novel molecular mechanism underlying the crosstalk between exosomal miRNAs and innate immune response in invertebrates.

The function of miR-7847 has not been reported previously, while miR-137 is regarded as an important regulator during tumorigenesis. For instance, miR-137 is reported to inhibit the proliferation of lung cancer cells by targeting Cdc42 and Cdk6 [40]. MiR-137 can also regulate the tumorigenicity of colon cancer stem cells through the inhibition of DCLK1 [41]. Prior to the present study, the roles of miR-137 and miR-7847 in invertebrates had remained unclear. The current study revealed that miR-137 and miR-7847 could suppress viral infection by promoting apoptosis. In addition, our data revealed that AIF was co-targeted by both miR-137 and miR-7847 in mud crabs. AIF is a mitochondrial FAD-dependent oxidoreductase protein that is involved in the regulation of oxidative phosphorylation [42], and is the first caspase-independent protein identified to be important in the mitochondrial pathway mediated apoptosis [43]. During the early stages of apoptosis, AIF is released from mitochondria and translocates to the nucleus [44], where it induces nuclear chromatin condensation, DNA fragmentation and cell death [45, 46]. When human alveolar epithelial cells (A549 cells) are challenged with influenza virus, AIF translocates from mitochondria to nucleus, resulting in apoptosis induction in response to the virus infection [47]. At present, the function of AIF in the immunoregulation of invertebrate has not been reported. Here, we found that AIF could inhibit WSSV infection by activating apoptosis of hemocytes in mud crabs. Moreover, AIF did not only translocate to the nucleus to induce DNA fragmentation, but was also found to competitively bind to HSP70 thereby disintegrating the HSP70-Bax complex, and freeing Bax from the mitochondria to activate mitochondrial apoptosis pathway. The present study therefore provides some novel insights into the invertebrate innate immune system and highlights potential preventive and therapeutic strategies for viral diseases.

In summary, our findings revealed the evolutionary conservation of exosomal regulatory pathway in animals. As a topical research area, studies relevant to exosomes have largely focused on higher organisms [48]. In invertebrates, exosomes have been reported only in *Drosophila*, and shown to be involved in the regulation of viral infection [49, 50] and miRNA biogenesis [51]. Thus, there is no enough evidence to conclude that exosome is a general regulatory approach in animals. In addition, exosomal miRNAs are also still unexplored in invertebrates, which means that there is an urgent need to characterize the existence and role of exosomal miRNAs. The current study is therefore the first to reveal the involvement of exosomal miRNAs in antiviral immune response of mud crabs, which shows a novel molecular mechanism of invertebrates resistance to pathogenic microbial infection.

## Materials and methods

### Ethics statement

The mud crabs used in this study were taken from a local crab farm (Niutianyang, Shantou, Guangdong, China). No specific permits were required for the described field studies, as the sampling locations were not privately owned or protected in any way. Furthermore these field studies did not involve endangered or protected species. The animals were processed according to "the Regulations for the Administration of Affairs Concerning Experimental Animals" established by the Guangdong Provincial Department of Science and Technology on the Use and Care of Animals.

### Mud crab culture and WSSV challenge

Healthy mud crabs, approximately 50 g each, were acclimated in the thanks filled with seawater at 25 ºC for 3 days before WSSV challenge. To ensure that the crabs were virus-free before the experiments, PCR analysis were performed via WSSV-specific primer (5'-TATTGTCTC TCCTGACGTAC-3' and 5'-CACATTCTTCACGAGTCTAC-3'). Then, 200 μL WSSV ($1\times10^{6}$ *cfu*/mL) was injected into the base of the fourth leg of each crab. At different time post-infection, hemolymph was collected from three randomly chosen crabs per group for further investigation.

### Isolation and analysis of exosomes

To isolate exosomes from mud crabs, hemolymph was centrifuged at 4 ºC, $300 \times g$ for 5 min and the sediments discarded. Next, the supernatant was further centrifuged at 4 ºC, $2,000 \times g$ for 30 min, followed by centrifugation of the supernatant at $10,000 \times g$ for 1 h at 4 ºC and finally centiguration of the collected supernatant at $130,000 \times g$ for 2 h at 4 ºC. The sediment was then collected after the last centrifugation and resuspended with 4.5 mL 0.95 M sucrose solution, and then 4.5 mL 2.0 M sucrose solution, and 4.5 mL 1.3 M sucrose solution were added to the 4.5 mL resuspended solution in a centrifuge tube before being centrifuged at 4 ºC, $200,000 \times g$ for 16 h. Finally, the solution between 1.3 M and 0.95 M sucrose solution was obtained and filtrated through 0.22 μm filters. The obtained exosomes were observed by Philips CM120 BioTwin transmission electron microscope (FEI Company, USA) and quantified by Nano-Sight NS300 (Malvern Instruments Ltd, UK).

### Exosomes tracing

For exosome-tracing experiments, the isolated exosomes were pre-treated with DiO (Beyotime, China), followed by injection of 200 μL of exosome solution ($1\times10^{8}$ vesicles/mL) into the base of the fourth leg of each crab. After 2 h post-injection, hemocytes were isolated and treated with DiI (Beyotime), followed by observation with confocal laser scanning microscopy TCS SP8 (Leica, Germany).

### Microarray analysis of exosomal miRNAs

Exosomal miRNAs microarray analysis was performed at Biomarker Technologies (Beijing, China), using Agilent Human miRNA 8*60 K V21.0 microarray (Agilent Technologies, USA). The NCBI BioProject database accession number is PRJNA600674. Gene Spring Software 12.6 (Agilent Technologies) was used for quantile normalization and data processing. Besides, Pearson's correlation analysis through Cluster 3.0 and TreeView software was used for Hierarchical clustering analysis of the differential expression of miRNAs.

## RNA interference assay

Based on the sequence of *Sp*-AIF (GenBank accession number MH393923.1) and *Sp*-HSP70 (GenBank accession number EU754021.1), the siRNA specifically targeted *Sp*-AIF or *Sp*-HSP70 gene was designed, generating AIF-siRNA (5'- UCUAAUUCUGCAUUGACUCUG UU -3') and HSP70-siRNA (5'- UCUUCAUAAGCACCAUAGAGGAGUU-3'). The siRNAs were synthesized using *in vitro* Transcription T7 Kit (TaKaRa, Dalian, China) according to the user's instructions. Then, 50 µg AIF-siRNA or HSP70-siRNA was injected into each mud crab respectively. At different time post siRNA injection, three mud crabs were randomly selected for each treatment and stored for further use.

## Quantification of mRNA with real-time PCR

The real-time quantitative PCR was conducted with the Premix Ex Taq (Takara, Japan) to quantify the mRNA level. Total RNA was extracted from hemocytes, followed by first-strand cDNA synthesis using PrimeScript RT Reagent Kit (Takara, Japan). Primers AIF-F (5'-AGC CATTGCCAGTCTTTGAT-3') and AIF-R (5'-GAACCCAGAAATCCTCCACC-3') was used to quantify the AIF mRNA transcript, while primers β-actin (β-actin-F, 5'-GCGGCAGTGGT CATCTCCT-3' and β-actin-R, 5'-GCCCTTCCTCACGCTATCCT-3') was used to quantify the internal control β-actin. Relative fold change of mRNA expression level of AIF was determined using the $2^{-\Delta\Delta Ct}$ algorithm [52].

## Quantification of miRNA with real-time PCR

Total RNA was extracted using MagMAX mirVana Total RNA Isolation Kit (Thermo Fisher Scientific, USA), followed by first-strand cDNA synthesis via PrimeScript II 1st Strand cDNA Synthesis Kit (Takara, Japan) using miR-137-primer (5'-GTCGTATCCAGTGCAGGGTCC GAGGTCACTGGATACGACACGTGTAT-3') and miR-7847-primer (5'- GTCGTATCCAG TGCAGGGTCCGAGGTCACTGGATACGACAATCCTCC-3'). Real-time PCR was carried out with the Premix Ex Taq (Takara, Japan) to quantify the expression level of miR-137 and miR-7847, U6 was used as control, the primers used were listed below. miR-137-F (5'- CGC CGTTATTGCTTGAGA-3') and miR-137-R (5'- TGCAGGGTCCGAGGTCACTG-3'), miR-7847-F (5'-CGCCGCTGGAGGAGTAGG-3') and miR-7847-R (5'- TGCAGGGTCCGAGGT CACTG-3'), U6-F (5'-CTCGCTTCGGCAGCACA-3') and U6-R (5'-AACGCTTCACGAAT TTGCGT-3').

## Analysis of WSSV copies by real-time PCR

The genomic DNA of WSSV-infected mud crab was extracted using a SQ tissue DNA (Omega Bio-tek, Norcross, GA, USA) according to the manufacturer's instruction. To detect WSSV copies in mud crab, real-time PCR analysis was carried out using Premix Ex Taq (probe qPCR) (Takara, Dalian, China). Real-time PCR was performed with WSSV-specific primers WSSV-RT1 (5'-TTGGTTTCATGCCCGAGATT-3') and WSSV-RT2 (5'-CCTTGGTCAGCC CCTTGA-3') and a TaqMan probe (5'-FAM-TGCTGCCGTCTCCAA-TAMRA-3') according to previous study [53]. The internal standard of real-time PCR was a DNA fragment of 1400 bp from the WSSV genome [54].

## Detection of apoptotic activity

In order to evaluate the apoptotic activity of mud crab, the Caspase 3/7 activity of hemocytes was determined with the Caspase-Glo 3/7 assay (Promega, USA). While the apoptosis rate was evaluated using FITC Annexin V Apoptosis Detection Kit I (BD Pharmingen, USA) according

to the manufacturer's instruction. And flow cytometry (Accuri C6 Plus, BD biosciences, USA) was used to analyze the apoptosis rate. Besides, the mitochondrial membrane potential, an indicator of the apoptotic activity in hemocytes, which were measured by Mitochondrial membrane potential assay kit with JC-1(Beyotime, China)following the protocols and finally analyzed by confocal microscope (ZEISS, Germany).

## Western blot analysis

The hemocytes of mud crab were homogenized with RIPA buffer (Beyotime Biotechnology, China) containing 1 mM phenylmethanesulfonyl fluoride (PMSF) and then centrifuged at 13,000×g for 5 min at 4 °C. Then the cell extracts mixed with 5 × SDS sample buffer were separated by 12% SDS-polyacrylamide gel electrophoresis and transferred onto a nitrocellulose membrane (Millipore, USA). Subsequently, the membrane was incubated in blocking buffer (Tris-buffered saline containing 0.1% Tween 20 (TBST) and 5% (W/V) nonfat dry milk) and further incubated with appropriate primary antibody at 4 °C. After washed with TBST, the membrane was incubated with horseradish peroxidase-conjugated secondary antibody (Bio-Rad, USA) for subsequent detection by ECL substrate (Thermo Scientific, USA).

## The target gene prediction of miRNA

Targetscan (http://www.targetscan.org) and miRanda (http://www.microrna.org/) were used to predict the target genes of miR-137 and miR-7847 by a commercial company (BioMarker, Beijing, China). And the overlapped target gene predicted by the two algorithms were the candidate target gene.

## Cell culture, transfection, and fluorescence assays

The *Drosophlia* Schneider 2 (S2) cells were cultured in Express Five serum-free medium (SFM) (Invitrogen, USA) at 27 °C. The EGFP-AIF-3'UTR or the mutant plasmids (100 ng/well) and the synthesized miR-137 or miR-7847 (50 nM/well) were co-transfected into S2 cells using with Cellfectin II Reagent (Invitrogen, USA) according to the manufacturer's protocol. At 48 h after co-transfection, the EGFP fluorescence in S2 cells was measured by a Flex Station II microplate reader (Molecular Devices, USA) at 490/ 510 nm of excitation/emission (Ex/ Em).

## The silencing or overexpression of miR-137 and miR-7847 in mud crab

Anti-microRNA oligonucleotide (AMOs) or miRNA mimic was injected at 30 μg/crab to knockdown or overexpress miRNAs in mud crab, AMO-miR-137 (5'- ACGTGTATTCT CAAGC**A**AT**A**A-3'), AMO-miR-7847 (5'-AATCCTCCTACTCC**T**CC**A**G-3'), miR-137 mimic (5'-TTATTGCTTGAGAATA**C**AC**G**T-3') and miR-7847 mimic (5'-CTGGAGGAGTAGG A**G**GA**T**T-3') were modified with 2'-O-methyl (OME) (bold letters) and phosphorothioate (the remaining nucleotides). All oligonucleotides were synthesized by Sangon Biotech (Shanghai, China). At different time points after the last injection, three mud crabs per treatment were collected for subsequent use.

## Fluorescence *in situ* hybridization

The hemocytes of mud crab were seeded onto the polysine-coated cover slips, fixed with 4% polyformaldehyde for 15 min at room temperature. After that, the cover slips were dehydrated in 70% ethanol overnight at 4˚C, followed by incubation with hybridization buffer [1× SSC (15 mM sodium citrate, 150 mM sodium chloride, pH 7.5), 10% (w/v) dextran sulfate, 25% (w/v)

formamide, 1× Denhardt's solution] containing 100 nM probe for 5 h at 37˚C. The miR-137 probe (5'-FAM- ACGTGTATTCTCAAGCAATAA-3'), miR-7847 probe (5'-FAM- AATCCT CCTACTCCTCCAG-3') and AIF probe (5'-Cy3-TCCATCTTCTGTACTCTTGACT-3') were used. Then the slips were washed with PBS for three times, and after that the hemocytes were stained with DAPI (4', 6-diamidino-2-phenylindole) (50 ng/mL) (Sigma, USA) for 5 min [55]. The images were captured using a CarlZeiss LSM710 system (Carl Zeiss, Germany).

## Statistical analysis

All data were subjected to one-way ANOVA analysis using Origin Pro 8.0, with $P < 0.01$ considered statistically significant. All assays were biologically repeated for three times.

## Author Contributions

**Data curation:** Yi Gong, Shengkang Li.

**Funding acquisition:** Yi Gong, Shengkang Li.

**Investigation:** Yi Gong, Tongtong Kong, Xin Ren, Jiao Chen, Shanmeng Lin.

**Methodology:** Yueling Zhang, Shengkang Li.

**Project administration:** Shengkang Li.

**Validation:** Yi Gong, Tongtong Kong, Xin Ren, Jiao Chen, Shanmeng Lin.

**Writing – original draft:** Yi Gong.

**Writing – review & editing:** Shengkang Li.

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
