## [Decision Letter · Decision Letter 0]

3 Apr 2020

Dear Dr. Li,

Thank you very much for submitting your manuscript "Exosome-mediated apoptosis pathway during WSSV infection in crustacean mud crab" for consideration at PLOS Pathogens. As with all papers reviewed by the journal, your manuscript was reviewed by members of the editorial board and by several independent reviewers. In light of the reviews (below this email), we would like to invite the resubmission of a significantly-revised version that takes into account the reviewers' comments.

We cannot make any decision about publication until we have seen the revised manuscript and your response to the reviewers' comments. Your revised manuscript is also likely to be sent to reviewers for further evaluation.

Sincerely,

Robert F. Kalejta

Associate Editor

PLOS Pathogens

Blossom Damania

Section Editor

PLOS Pathogens

Kasturi Haldar

Editor-in-Chief

PLOS Pathogens

orcid.org/0000-0001-5065-158X

Michael Malim

Editor-in-Chief

PLOS Pathogens

orcid.org/0000-0002-7699-2064

Reviewer's Responses to Questions

**Part I - Summary**

Reviewer #1: In this manuscript, Gong and collaborators revealed the involvement of exosomal miRNAs in antiviral immune response of crustacean mud crab. In invertebrates, the exosome relevant researches are quite limited. This manuscript firstly identifies the exosomes in crustaceans. In general, this is a valuable contribution to our understanding of exosome-mediated immune pathway of crustaceans and the host defense against the invading virus. Overall, the results described in the manuscript looks very convincing and some results sound interesting to the readers in the relevant fields. The authors are very proficient at the experimental techniques and relatively reasonable at experimental design. However, some issues need to be addressed.

Reviewer #2: The current manuscript titled “Exosome-mediated apoptosis pathway during WSSV infection in crustacean mud crab” describes how exosomes released from WSSV-injected mud crabs could suppress viral invasion by inducing apoptosis of hemocytes. Using WB analysis of exosome markers (CD9 and TSG101) and cytoplasmic marker (calnexin), the authors have found that Caspase 3/7 activity was significantly increased in the exosome-WSSV and WSSV co-injected group compared to the control group. Here, the controls included the apoptosis inducer cycloheximide and apoptosis inhibitor Z-VAD-FMK for their effects on WSSV proliferation in hemocytes of mud crab. Results showed lower WSSV copy number in the cycloheximide and WSSV-injected group, but significantly higher WSSV copy number in the Z-VAD-FMK and WSSV-injected group. Finally, in an attempt to reveal the roles of miR-137 and miR-7847 in apoptosis, mud crabs were injected with AMO-miR-137 and AMO-miR-7847, and Flow analysis revealed that both AMO-miR-137 and AMO-miR-7847 induced higher apoptosis compared to the controls.

**Part II – Major Issues: Key Experiments Required for Acceptance**

Reviewer #1: No.

Reviewer #2: Overall, this is an interesting manuscript and there are some novel findings that is validated in an animal model. The concerns include:

1. It is not clear what is the ratio of EVs to Cells, i.e., Is it 1:1010, 1:1000, 1:10000 to score for an effect?

2. Data in Figure 2D requires at least multiple WBs to validate the results with confidence

3. For mir137 to work in cells, they require high levels of miRNA machinery components. What are the levels of Ago2, Dicer, TRBP, etc in the recipient cells?

4. What percent of HSP70 binds to AIF vs Bax?

**Part III – Minor Issues: Editorial and Data Presentation Modifications**

Reviewer #1: 1) The manuscript contains some ill sentences and grammar errors. Addressing these concerns would help them to improve their manuscript. For example, the Latin name spelling should be Scylla paramamosain, not Scally paramamosain.

2) In references section, article titles should only be capitalized on the first word, not the first letter of each word.

3) The protocols of exosomes isolation should be described in detail. At present, there is no mature instruction for the exosomes isolation in crustaceans.

4) The authors found that two exosomal miRNAs (miR-137 and miR-7847) were both involved in antiviral immune regulation in mud crab. It is necessary to explore whether there exist primary or secondary relationships during the regulatory process.

5) The authors only detect the interactions between miRNAs and AIF, while the relationship between exosomes and AIF is not studied. Thus, the effect of exosomes challenge on AIF expression should also be investigated.

6) It is found from the data that AIF could interact with HSP70, while HSP70 could interact with Bax. Thus, whether AIF could directly interact with Bax should be clarified in detail accordingly.

Reviewer #2: None

PLOS authors have the option to publish the peer review history of their article (what does this mean?). If published, this will include your full peer review and any attached files.

Reviewer #1: No

Reviewer #2: No
---

## [Editor Report · Decision Letter 1]

29 Apr 2020

Dear Dr. Li,

We are pleased to inform you that your manuscript 'Exosome-mediated apoptosis pathway during WSSV infection in crustacean mud crab' has been provisionally accepted for publication in PLOS Pathogens.

Best regards,

Robert F. Kalejta

Associate Editor

PLOS Pathogens

Blossom Damania

Section Editor

PLOS Pathogens

Kasturi Haldar

Editor-in-Chief

PLOS Pathogens

orcid.org/0000-0001-5065-158X

Michael Malim

Editor-in-Chief

PLOS Pathogens

orcid.org/0000-0002-7699-2064
---

## [Editor Report · Acceptance letter]

13 May 2020

Dear Dr. Li,

We are delighted to inform you that your manuscript, "Exosome-mediated apoptosis pathway during WSSV infection in crustacean mud crab," has been formally accepted for publication in PLOS Pathogens.

Best regards,

Kasturi Haldar

Editor-in-Chief

PLOS Pathogens

orcid.org/0000-0001-5065-158X

Michael Malim

Editor-in-Chief

PLOS Pathogens

orcid.org/0000-0002-7699-2064